# Effects of Once-Weekly Semaglutide on Cardiovascular Risk Factors and Metabolic Dysfunction-Associated Steatotic Liver Disease in Japanese Patients with Type 2 Diabetes: A Retrospective Longitudinal Study Based on Real-World Data

**DOI:** 10.3390/biomedicines12051001

**Published:** 2024-05-02

**Authors:** Hisayuki Katsuyama, Mariko Hakoshima, Emika Kaji, Masaaki Mino, Eiji Kakazu, Sakura Iida, Hiroki Adachi, Tatsuya Kanto, Hidekatsu Yanai

**Affiliations:** 1Department of Diabetes, Endocrinology and Metabolism, National Center for Global Health and Medicine Kohnodai Hospital, 1-7-1 Kohnodai, Ichikawa 272-8516, Chiba, Japan; d-hakoshima@hospk.ncgm.go.jp (M.H.); d-21kaji@hospk.ncgm.go.jp (E.K.); d-20iida@hospk.ncgm.go.jp (S.I.); dadachidm@hospk.ncgm.go.jp (H.A.); dyanai@hospk.ncgm.go.jp (H.Y.); 2Department of Gastroenterology and Hepatology, National Center for Global Health and Medicine Kohnodai Hospital, 1-7-1 Kohnodai, Ichikawa 272-8516, Chiba, Japan; d-21mino@hospk.ncgm.go.jp (M.M.); d-21kakazu@hospk.ncgm.go.jp (E.K.); kantot@hospk.ncgm.go.jp (T.K.); 3Department of Liver Diseases, The Research Center for Hepatitis and Immunology, National Center for Global Health and Medicine, 1-21-1 Toyama, Shinjuku-ku, Tokyo 162-8655, Japan

**Keywords:** GLP-1 receptor agonist, semaglutide, type 2 diabetes, dyslipidemia, MASLD, T2D

## Abstract

Once-weekly semaglutide is a widely used glucagon-like peptide-1 receptor agonist (GLP-1RA) used for the treatment of type 2 diabetes (T2D). In clinical trials, semaglutide improved glycemic control and obesity, and reduced major cardiovascular events. However, the reports are limited on its real-world efficacy relating to various metabolic factors such as dyslipidemia or metabolic dysfunction-associated steatotic liver disease (MASLD) in Asian patients with T2D. In our retrospective longitudinal study, we selected patients with T2D who were given once-weekly semaglutide and compared metabolic parameters before and after the start of semaglutide. Seventy-five patients were eligible. HbA1c decreased significantly, by 0.7–0.9%, and body weight by 1.4–1.7 kg during the semaglutide treatment. Non-HDL cholesterol decreased significantly at 3, 6 and 12 months after the initiation of semaglutide; LDL cholesterol decreased at 3 and 6 months; and HDL cholesterol increased at 12 months. The effects on body weight, HbA1c and lipid profile were pronounced in patients who were given semaglutide as a first GLP-1RA (GLP-1R naïve), whereas improvements in HbA1c were also observed in patients who were given semaglutide after being switched from other GLP-1RAs. During a 12-month semaglutide treatment, the hepatic steatosis index (HSI) tended to decrease. Moreover, a significant decrease in the AST-to-platelet ratio index (APRI) was observed in GLP-1RA naïve patients. Our real-world study confirmed the beneficial effects of once-weekly semaglutide, namely, improved body weight, glycemic control and atherogenic lipid profile. The beneficial effects on MASLD were also suggested.

## 1. Introduction

The global prevalence of diabetes is estimated to be 463 million people, rising to 578 million by 2030 [1], and more than 60% of people with diabetes live in Asia [2]. Obesity is a major driver of the epidemic of type 2 diabetes (T2D) [3], and the presence of T2D is associated with an elevated risk of cardiovascular (CV) diseases in the Asian population [4,5]. Interventions against multiple CV risk factors, such as diabetes, dyslipidemia, hypertension and obesity, can play a key role in the prevention of CV diseases [6].

Obesity and T2D also relate to the development and progression of metabolic dysfunction-associated steatotic liver disease (MASLD), which comprises simple steatosis, metabolic dysfunction-associated steatohepatitis (MASH), fibrosis/cirrhosis, and hepatocellular carcinoma [7]. The progression of MASLD is not only a risk factor for liver-related death but also for CV diseases and T2D comorbidities such as chronic kidney disease [8,9]. Preventing MASLD progression is considered one of the important factors in diabetic treatments.

Glucagon-like peptide-1 (GLP-1) is an intestinal hormone that stimulates insulin secretion and inhibits glucagon secretion from pancreatic islets. GLP-1 receptor agonists (GLP-1RAs) have been demonstrated to improve glycemic control and reduce body weight in various clinical trials [10,11]. Our previous study of once-weekly dulaglutide showed positive effects of GLP-1RAs on various metabolic factors such as dyslipidemia, blood pressure and liver dysfunction [12]. Once-weekly semaglutide is a subcutaneous GLP-1RA, and widely used for the treatment of T2D. In a randomized controlled study, the use of once-weekly semaglutide was associated with a lower incidence of CV events in patients with T2D who were at high CV risk [13]. Comparisons of previous clinical studies suggested that once-weekly semaglutide could have stronger effects in lowering glucose levels or body weight [14]. Nevertheless, most of the insights on semaglutide are based on clinical trials conducted mainly in Western regions, and only a limited number of reports have studied the real-world effects of once-weekly semaglutide on various metabolic factors in Asian patients with T2D, a group characterized by relatively lower body mass index (BMI) and a more severe insulin secretion deficiency [15]. This study aimed to examine the real-world efficacy of once-weekly semaglutide on various CV risk factors and MASLD in Japanese patients with T2D.

## 2. Materials and Methods

### 2.1. Study Population

The retrospective longitudinal study based on medical charts was conducted at the National Center for Global Health and Medicine, Kohnodai Hospital, Japan. We enrolled patients with T2D who had been prescribed once-weekly semaglutide for at least 3 months or longer between 1 June 2020 and 30 June 2022. We excluded the patients who did not visit our hospital regularly. The patients for whom semaglutide prescription was suspended within 3 months were also excluded from the analysis.

### 2.2. Data Collection

We collected the relevant data on various metabolic factors, including results from blood tests and urine tests and anthropometric parameters, and compared the data before and after the initiation of semaglutide treatment. Information on concomitant treatments was also collected from medical charts. Body weight, height, waist circumference and blood pressure were measured according to the clinical standards. Body mass index (BMI) was calculated by dividing body weight in kilograms by body height squared in meters. The measurements of serum hemoglobin A1c (HbA1c), total cholesterol (TC), triglyceride (TG), creatinine, and uric acid were performed using enzymatic assays. The hexokinase method was indicated for the evaluation of plasma glucose. A direct method was used for the measurements of serum low-density lipoprotein cholesterol (LDL-C) and HDL-C. Serum transaminases, including aspartate aminotransferase (AST), alanine aminotransferase (ALT) and γ-glutamyl transferase (GGTP) were measured using the Japan Society of Clinical Chemistry transferable method. A turbidimetric immunoassay was used for the measurement of urinary albumin, and the albumin-to-creatinine ratio (UACR) was calculated. UACR stage (A1–A3) was determined based on the Kidney Disease Improving Global Outcomes (KDIGO) risk categories [16]. The estimated glomerular filtration rate (eGFR) was calculated by age and serum creatinine based on the estimation equation for Japanese patients [17]. Non-HDL-C was calculated by subtracting HD-C from TC. The hepatic steatosis index (HSI) was calculated using the following formula: 8 × (ALT/AST) + BMI + (2, if diabetes mellitus) + (2, if female) [18]. The AST-to-platelet ratio index (APRI) and fibrosis -4 (FIB-4) index are markers for hepatic fibrosis. The APRI was calculated as follows: AST (IU/L)/Upper limit of the normal range of AST: 40 (IU/L)/Platelet count (10^9^/L) × 100 [19]. FIB-4 index was calculated as follows: (age × AST)/(platelet counts (×10^9^/L) × (ALT)^1/2^ [20,21].

### 2.3. Statistical Analysis

Data obtained in this study were tested for normality using the Shapiro–Wilk test. Comparisons of the variables with normal distribution were analyzed by paired *t*-tests. Variables without normal distribution were compared using the Wilcoxon signed-rank test. Spearman’s rank correlation coefficient was used to determine the correlations between the parameters. Missing data were excluded from analyses. All data are expressed as mean ± SD, and *p* < 0.05 was considered to be statistically significant. We used SPSS version 29 (IBM Corp, Armonk, NY, USA) for statistical analysis.

## 3. Results

### 3.1. Characteristics of Patients

During the observation period, 83 patients were prescribed once-weekly semaglutide. In five patients, semaglutide prescription was suspended, due to either injection-site pain (*n* = 3), gastrointestinal symptoms (*n* = 1) or unavailability due to supply shortage (*n* = 1). Three patients were excluded due to lack of regular visits to our hospital. Thus, we examined 75 patients in this study. All patients were under standard care, including recommendations for diet and physical activities according to the guidelines of the Japan Diabetes Society.

Table 1 shows the baseline characteristics of the patients. The mean age of the patients was 55.8 ± 13.3 years, and the mean BMI was 31.4 ± 5.2 kg/m^2^. A total of 36 patients were prescribed semaglutide as a first GLP-1RA (GLP-1RA naïve). Thirty-nine patients were given semaglutide switched from other GLP-1RAs (thirty patients from once-weekly dulaglutide 0.75mg, five patients from once-daily liraglutide 0.9 mg, and one patient from oral semaglutide 3mg). In 16 patients, DPP-4 inhibitors were switched to semaglutide. Among hypoglycemic agents, for the most part, SGLT2 inhibitors (SGLT2is) were used (91%), followed by metformin (69%) and Thiazolidinedione (24%). Insulin was used in 13 patients (17%). Angiotensin II receptor blockers (ARB) were used in 38 patients (51%) and calcium channel blockers in 35 patients (47%). Statins were given for 44 patients (59%). Antiplatelet drugs were prescribed in 14 patients (19%).

The initial dose of once-weekly semaglutide was 0.25 mg in 68 patients. In seven patients who were given semaglutide switched from other GLP1Ras, semaglutide was started at 0.5 mg. At 3 months after the initiation of semaglutide, the doses of semaglutide were 0.25 mg in sixteen patients, 0.5 mg in fifty-seven patients and 1.0 mg in two patients. At 6 months, seven patients received semaglutide at 0.25 mg, forty-one patients at 0.5 mg and six patients at 1.0 mg. At 12 months, semaglutide was given at 0.25 mg in three patients, 0.5 mg in twenty-seven patients and 1.0 mg in ten patients.

### 3.2. Changes in Metabolic Parameters during Semaglutide Treatment

Table 2 provides the changes in metabolic parameters during semaglutide treatments. Overall, HbA1c decreased significantly, by 0.7–0.9%, and body weight by 1.4–1.7 kg. Improvements in lipid profile were also observed. There were significant decreases in non-HDL-C at 3, 6 and 12 months, LDL-C at 3 and 6 months and TG at 12 months. HDL-C increased at 12 months. AST and ALT decreased at 3, 6 and 12 months and GGTP decreased at 3 months. There were no significant changes in eGFR and UACR. The mean eGFR slope was calculated as 0.55 mL/min/1.73 m^2^/year.

Figure 1 shows the changes in body weight, HbA1c, AST and non-HDL-C in patients who were continuously prescribed semaglutide for 12 months. Body weight, HbA1c and AST decreased at 3 months after the initiation of semaglutide and maintained these significant differences until the 12-month point.

### 3.3. Changes in Metabolic Parameters in GLP-1 RA Naïve Patients or Patients Given Semaglutide Switched from Other GLP-1RAs

We compared the patients who were given semaglutide as a first GLP-1RA (Group A) with those switched from other GLP-1RAs (Group B) at 12 months after the initiation of semaglutide (Table 3). In group B, there were significant decreases in HbA1c, by 0.5%. Body weight and BMI also tended to decrease. Nevertheless, there were no significant changes in liver transaminases and serum lipids, except for HDL-C. In group A, a significant decrease was observed in HbA1c, of 1.3%, and there was the same tendency in body weight, by 2.0 kg. There were significant improvements in non-HDL-C at 12 months. Moreover, AST, ALT and GGTP decreased significantly. Overall, the results at 3 and 6 months after the initiation of semaglutide resembled were similar to those at 12 months (Appendix A).

### 3.4. Changes in MASLD Indices during 12-Month Semaglutide Treatment

Table 4 provides the changes in MASLD indices at 12 months after the initiation of semaglutide treatment. There was a tendency towards a decrease in HSI, whereas the APRI and FIB4-index showed no significant changes (Table 4a).

In patients given semaglutide switched from other GLP-1RAs (Group B), there were no differences in the HSI, APRI or FIB-4 index (Table 4 (b)). In GLP-1RA naïve patients (Group A), the APRI was improved significantly, and the HSI and FIB-4 index also tended to decrease.

To elucidate the effects of semaglutide in patients in the presence of liver damage, we divided the patients into groups according to the baseline values of ALT levels. In patients with a higher degree of liver damage (baseline ALT ≥ 30 IU/L), there was a significant decrease in the APRI and the FIB-4 index, whereas the APRI and FIB-4 index increased significantly in patients with a baseline ALT < 30 (Table 4c).

We also divided the patients into groups according to baseline LDL-C levels (Table 4d). The HSI showed a significant decrease only in patients with higher LDL-C levels (Baseline LDL-C ≥ 100 mg/dL). Moreover, we examined the changes in MASLD indices in patients divided by concomitant use of statins and ARB (Table 4e,f). The HSI decreased significantly in patients with statins or ARB, whereas there were no significant changes in the APRI or FIB-4 index.

### 3.5. Correlations between the Baseline and the Changes in MASLD Indices

Figure 2 shows the correlations between the baseline values and the changes in MASLD indices during 12-month semaglutide treatment. Significant correlations were found between the baseline values and changes in HSI (R = −0.332, *p* = 0.036), FIB-4 index (R = −0.333, *p* = 0.047) and APRI (R = −0.417, *p* = 0.007).

### 3.6. Correlations among the Changes in Metabolic Parameters

Table 5 provides the correlations between the changes in metabolic parameters during the 12 months of semaglutide treatment. The changes in the HSI were correlated with the changes in BMI (R = 0.536, *p* = 0.002), but not correlated with the changes in HbA1c (R = −0.002, *p* = 0.991). There were significant correlations between the changes in HbA1c and the changes in the FIB-4 index (R = 0.511, *p* = 0.001) and the APRI (R = 0.494, *p* = 0.001), whereas the changes in BMI were not correlated with the changes in the FIB-4 index and APRI.

### 3.7. Changes in UACR Stages during 12-Month Semaglutide Treatment

Table 6 shows the changes in the UACR stage between baseline and 12 months after the initiation of semaglutide treatment. Remission in the UACR stage was observed in five patients (A2–A1: 3 patients, A3–A2: 2 patients), whereas deterioration was observed in one patient.

## 4. Discussion

In this real-world study, treatment with once-weekly semaglutide reduced body weight and improved hyperglycemia and atherogenic dyslipidemia in Japanese patients with T2D. It was also suggested that semaglutide had beneficial effects on preventing the progression of MASLD in high-risk patients.

Clinical trials already provided consistent evidence of the effects of the weekly semaglutide as to improvements in obesity and glycemic control. In Japanese patients with T2D, semaglutide treatment at a dose of 0.5 mg or 1.0 mg reduced HbA1c by 1.7–2.2% and body weight by 1.4 kg–3.9 kg [10,11]. Our real-world data confirmed the improvements in glycemic control and body weight associated with once-weekly semaglutide. However, the degrees of the changes in HbA1c and body weight were smaller. Compared to the clinical trials, most of the patients in our study had already been treated with other anti-diabetic agents such as SGLT2is or metformin. Moreover, our data included the patients who had been given other GLP-1RAs before the initiation of semaglutide. As shown in Table 3, the changes in HbA1c and body weight were greater in GLP-1RA naïve patients. A real-world study examining the effects of once-weekly semaglutide in Japanese patients reported changes in HbA1c and body weight similar to those observed in the present study [22]. Furthermore, our study demonstrated the multifactorial benefits of semaglutide against CV risk factors.

Once-weekly semaglutide has already been suggested to have stronger effects on improving glycemic control and obesity, compared with other GLP-1Ras, in a comparison of clinical trials [14]. Direct comparisons of once-weekly semaglutide and dulaglutide revealed the pronounced glucose-lowering and body-mass-index-lowering effects of semaglutide in Japanese patients with T2D [23,24]. Switching from liraglutide to once-weekly semaglutide also resulted in significant improvements in HbA1c and body weight, despite no significant changes being observed in patients given dulaglutide who had switched from liraglutide [25]. Our results also revealed a significant improvement in HbA1c as well as a tendency towards a decrease in body weight, even in patients who had been given other GLP-1RAs before the initiation of once-weekly semaglutide, which confirmed the stronger effects of once-weekly semaglutide on glycemia and body weight in the real world. Compared with the average BMI of around 24 kg/m^2^ [26], the patients in our study had higher BMI and coexisting metabolic disorders, including dyslipidemia and hypertension, suggesting that in clinical practice, once-weekly semaglutide was preferably prescribed for obese patients with a higher risk for CV diseases.

Insulin resistance is a common feature of T2D, dyslipidemia and MASLD. MASLD is characterized by the liver’s accumulation of TG, which is synthesized from fatty acyl coenzyme A (CoA). The concentration of fatty acyl CoA in the liver depends on the balance of the formation and utilization of free fatty acid. When the formation of FFAs from circulating FFAs, de novo lipogenesis, lipoprotein uptake and TG breakdown exceed lipid synthesis and oxidation, the fatty acyl CoA concentration rises, resulting in the accumulation of TG. Insulin resistance in adipose tissue activates hormone-sensitive lipase, which, in turn, promotes lipolysis and the release of FFA and proinflammatory cytokines which aggravate insulin resistance [7,27,28]. Circulating FFA levels are further augmented by dietary lipids, resulting in an increased influx of FFA into the liver, which, in turn, inhibits the degradation of apolipoprotein B 100 and promotes TG-rich, very low-density lipoprotein (VLDL) secretion, weheras the TG export is insufficient to normalize the hepatic TG content. De novo lipogenesis in the liver is also elevated by hyperinsulinemia resulting from insulin resistance, leading to further production of FFA and VLDL. Lipid overload in the liver increases mitochondrial beta-oxidation and TCA cycle activity, which can potentially induce oxidative stress, promoting liver damage and the progression of fibrosis. GLP-1RAs induce postprandial insulin secretion and improve insulin resistance, which inhibits lipolysis and FFA release in adipose tissue. The reduction of FFA release to the bloodstream results in decreased FFA influx to the liver. GLP-1RAs can also inhibit hepatic de novo lipogenesis and VLDL-TG secretion [29]. These effects of GLP-1RAs can lead to a reduction in hepatic TG content and improvements in atherogenic lipid profiles.

We used the HSI, APRI and FIB-4 index for the evaluation of MASLD. A previous study revealed the usefulness of the HSI for the detection of non-alcoholic fatty liver disease (NAFLD) [18]. It was also reported that the APRI and FIB-4 index were associated with the degree of hepatic fibrosis and the outcome of NAFLD [30,31,32,33]. Although the fatty liver index is also frequently used for the assessments of hepatic steatosis [34], it was impossible to use it for our study, since the regular measurements of waist circumference were not common in clinical practice in Japan. It should also be noted that there are few reports assessing the association between these indices and the severity or outcome of newly defined MASLD.

In our study, once-weekly semaglutide reduced the hepatic steatosis index. The attenuation of liver fat accumulation by semaglutide was also reported in a study using magnetic resonance imaging [35]. There was also a significant improvement in the APRI and the same tendency was seen in the FIB-4 index in GLP-1RA naïve patients in our study, suggesting the possible effects of once-weekly semaglutide, not only on hepatic steatosis, but also on hepatic fibrosis. The associations between the baseline values and changes in the APRI and FIB-4 index suggested that semaglutide can improve hepatic fibrosis in patients with advanced fibrosis at baseline. A randomized controlled study reported that 72 weeks of semaglutide yielded more patients with MASH resolution compared with placebo, nevertheless, there was no difference in the improvement of the fibrosis stage [36]. Interestingly, significant associations were observed between the changes in HbA1c and the changes in the APRI or FIB-4 index in our study, suggesting that enhanced insulin action may be a common mechanism in improving glycemic control and preventing the development of liver fibrosis.

SGLT2is, metformin, pioglitazone, statins and ARB have been reported to have beneficial effects on the progression of MASLD [37,38,39,40,41,42,43]. Thus, our results might be influenced by these concomitant agents. In our sub-analysis, the HSI decreased significantly only in patients with statins or ARB but not in patients without these drugs. According to a previous report, the use of a statin prevented the progression of MASLD to MASH but did not reduce liver fat accumulation [41]. It was also reported that ARB contributed to preventing the progression of MASLD to MASH [42,43]. Furthermore, an animal study showed that ARB could reduce liver fat accumulation [44]. Thus, the improvements in the HSI in our study could have been especially influenced by the use of ARB. Nevertheless, the APRI tended to decrease regardless of concomitant use of statins or ARB, which suggested the effects of semaglutide on the prevention of the progression of fibrosis in the liver. On the other hand, we could not compare the changes in MASLD indices in patients with or without SGLT2is and metformin, since most patients took them. Our results should be further examined in patients without these drugs.

Insulin resistance reduces the activity of lipoprotein lipase (LPL), leading to increases in intermediate-density lipoprotein (IDL) and VLDL and decreases in HDL concentration. Semaglutide can stimulate insulin action, reduce the influx of FFA to the liver and enhance LPL activities, leading to improvements in the lipoprotein profile, such as reductions in IDL and VLDL, which could be observed as siginificant decreases in non-HDL-C in our study. Previous studies also reported significant improvements in non-HDL-C, LDL-C, HDL-C and TG [45,46], findings which were consistent with our results. Improvements in the atherogenic lipid profile can assist in the protective role played by semaglutide as to CV diseases, which was demonstrated in the SUSTAIN 6 trial [14].

Although there were no significant differences in UACR during semaglutide treatment, remission in the UACR stage was observed in several patients. Previous studies also reported that once-weekly semaglutide reduced UACR and suggested the renoprotective effects of semaglutide [47,48]. The limited availability of data might have influenced our results.

In our study, 91% of the patients also took SGLT2 inhibitors, which have been reported to improve glycemia, lipid profile and obesity, as well as MASLD [37,49]. Adding once-weekly semaglutide to SGLT2 inhibitors resulted in greater reductions in body weight and HbA1c and was generally well tolerated [50]. A meta-analysis also revealed that a combination therapy comprising SGLT2i and GLP-1RA had beneficial effects on the progression of MASLD, and once-weekly semaglutide had a greatest advantage compared with other GLP-1RA [51].

Our study has several limitations. First, due to the retrospective design with the limited number of patients, our study could not avoid possible influences by various confounding factors. Second, although all patients were recommended to comply with appropriate diet and physical activity according to Japanese clinical standards, we could not take diet and physical activities into account due to difficulties in obtaining sufficient data. Third, our study lacked information on the diabetic duration and the state of coexisting diabetic complications. Lastly, our study lacked histological evaluations of the liver. Further study will be needed with the larger number of patients as well as a control group.

Despite these limitations, our study confirmed the results of previous clinical trials showing the effects of once-weekly semaglutide on body weight, HbA1c and atherogenic lipid profiles, which can be generalized to real-world patients with T2D.

## 5. Conclusions

Our real-world study showed that once-weekly semaglutide improved body weight, glycemic control and atherogenic lipidemia in Japanese patients with T2D. Moreover, it was suggested that semaglutide reduces hepatic fat content and has a preventative role against the progression of MASLD.

## Figures and Tables

**Figure 1 biomedicines-12-01001-f001:**
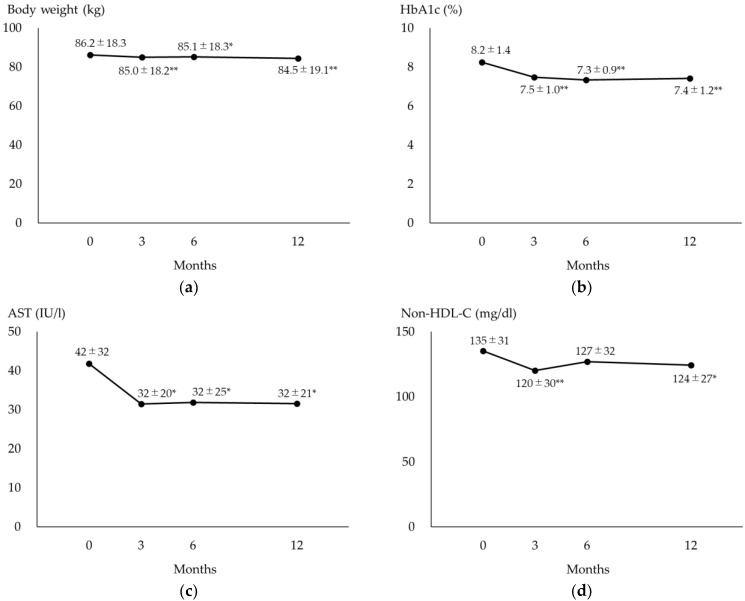
Changes in measured values in patients who were continuously prescribed semaglutide for 12 months. (**a**) The changes in body weight during the 12-month semaglutide treatment. (**b**) The changes in HbA1c during the 12-month semaglutide treatment. (**c**) The changes in AST during the 12-month semaglutide treatment. (**d**) The changes in non-HDL-C during the 12-month semaglutide treatment. * *p* < 0.05 vs. baseline, ** *p* < 0.01 vs. baseline. Values show mean ± SD. AST, aspartate aminotransferase; HbA1c, hemoglobin A1c; HDL-C, high-density lipoprotein cholesterol.

**Figure 2 biomedicines-12-01001-f002:**
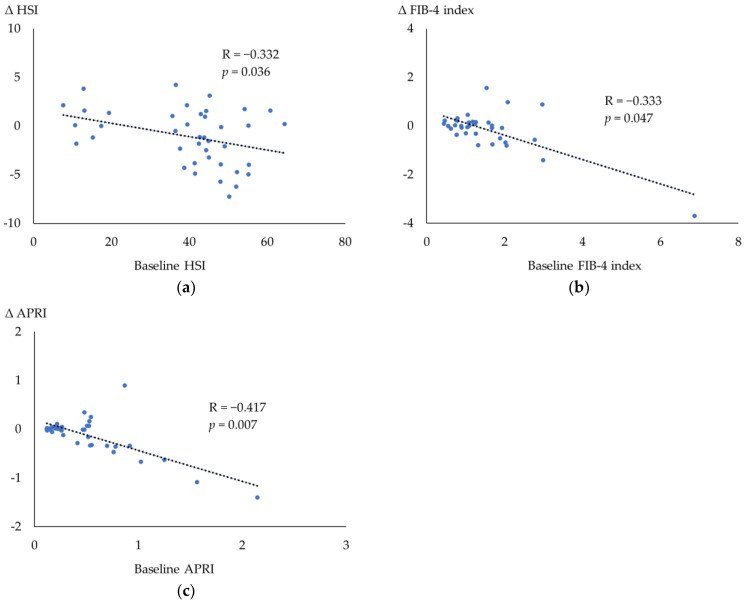
Spearman’s rank correlation coefficient among the parameters during the 12 months after the initiation of semaglutide. (**a**) The correlation between the changes in the HSI and the baseline HSI. (**b**) The correlation between the changes in the FIB-4 index and the baseline FIB-4 index. (**c**) The correlation between the changes in the APRI and the baseline. APRI. APRI, AST-to-platelet ratio index; AST, aspartate aminotransferase; FIB-4 index, fibrosis-4 index; HSI, hepatic steatosis index.

**Table 1 biomedicines-12-01001-t001:** The baseline characteristics of the patients (n = 75).

Age (year)	55.8 ± 13.3
Gender (M/F)	41/34
Body height (cm)	164.3 ± 7.4
Body weight (kg)	84.3 ± 16.2
BMI (kg/m^2^)	31.4 ± 5.2
GLP-1RA naïve Switch from other GLP-1RAs Switch from DPP-4 inhibitors	36 (48%) 39 (52%) 16 (21%)
Medications at baseline	
Insulin	13 (17%)
Metformin	52 (69%)
Sulfonylurea	8 (11%)
Glinides	8 (11%)
Thiazolidinedione	18 (24%)
Alpha-glucosidase inhibitors	8 (11%)
SGLT2 inhibitors	68 (91%)
Angiotensin II receptor blockers	38 (51%)
Calcium channel blockers	35 (47%)
Diuretics	8 (11%)
Beta blockers	8 (11%)
Statins	44 (59%)
Ezetimibe	15 (20%)
PCSK9 inhibitors	1 (1%)
Fibrates	17 (23%)
Antiplatelet drugs	14 (19%)

Values show mean ± SD. BMI, body mass index; DPP-4, dipeptidyl peptidase-4; GLP-1RA, glucagon-like peptide-1 receptor agonist; PCSK9, proprotein convertase subtilisin/kexin type 9; SGLT2, sodium-glucose transport protein 2.

**Table 2 biomedicines-12-01001-t002:** The changes in metabolic parameters during the semaglutide treatment in all patients.

**(a) The changes in metabolic parameters 3 months after the initiation of semaglutide treatment (n = 75).**				
	**n**	**Baseline**	**3 Months**	** *p* **
Body weight (kg)	64	84.6 ± 15.5	83.2 ± 15.1	0.004
BMI (kg/m^2^)	64	31.5 ± 5.2	30.9 ± 5.0	<0.001
Systolic blood pressure (mmHg)	65	133 ± 14	130 ± 14	0.077
Diastolic blood pressure (mmHg)	65	79 ± 11	79 ± 12	0.943
Plasma glucose (mg/dL)	73	173 ± 56	163 ± 49	0.219
HbA1c (%)	73	8.2 ± 1.3	7.5 ± 1.1	<0.001
Albumin (g/dL)	68	4.10 ± 0.56	4.19 ± 0.44	0.091
AST (IU/L)	75	37 ± 30	29 ± 20	<0.001
ALT (IU/L)	75	48 ± 37	39 ± 30	<0.001
GGTP (IU/L)	72	61 ± 61	51 ± 61	<0.001
TC (mg/dL)	69	178 ± 39	166 ± 31	0.002
HDL-C (mg/dL)	71	46 ± 10	47 ± 11	0.921
LDL-C (mg/dL)	62	95 ± 27	88 ± 24	0.029
TG (mg/dL)	71	225 ± 149	204 ± 135	0.203
TG/HDL-C	71	5.4 ± 4.2	5.1 ±5.0	0.249
Non-HDL-C (mg/dL)	68	130 ± 38	120 ± 31	<0.001
Creatinine (mg/dL)	75	0.80 ± 0.32	0.84 ± 0.53	0.515
eGFR (mL/min/1.73 m^2^)	75	79 ± 31	79 ± 29	0.680
Uric acid (mg/dL)	66	5.8 ± 1.6	5.6 ± 1.7	0.136
Hemoglobin (g/dL)	75	14.5 ± 1.7	14.5 ± 1.6	0.980
Platelet (×10^4^/μL)	75	24.1 ± 7.2	24.7 ± 8.0	0.093
UACR (mg/g Cre)	38	163 ± 294	114 ± 221	0.099
**(b) The changes in metabolic parameters 6 months after the initiation of semaglutide treatment (n = 54).**				
	**n**	**Baseline**	**6 Months**	** *p* **
Body weight (kg)	50	84.3 ± 16.2	82.6 ± 15.8	0.012
BMI (kg/m^2^)	50	31.4 ± 5.1	30.8 ± 5.3	0.010
Systolic blood pressure (mmHg)	51	132 ± 13	133 ± 17	0.720
Diastolic blood pressure (mmHg)	51	78 ± 10	79 ± 10	0.321
Plasma glucose (mg/dL)	54	177 ± 58	162 ± 48	0.149
HbA1c (%)	54	8.3 ± 1.4	7.4 ± 1.0	<0.001
Albumin (g/dL)	50	4.11 ± 0.52	4.17 ± 0.44	0.232
AST (IU/L)	54	38 ± 29	31 ± 22	0.009
ALT (IU/L)	54	50 ± 37	39 ± 28	0.007
GGTP (IU/L)	52	60 ± 60	51 ± 63	0.002
TC (mg/dL)	49	179 ± 42	168 ± 29	0.083
HDL-C (mg/dL)	53	46 ± 10	47 ± 12	0.329
LDL-C (mg/dL)	47	97 ± 30	89 ± 21	0.025
TG (mg/dL)	53	227 ± 137	213 ± 190	0.061
TG/HDL-C	53	5.5 ± 4.0	5.6 ± 6.9	0.065
Non-HDL-C (mg/dL)	50	133 ± 43	120 ± 31	0.040
Creatinine (mg/dL)	54	0.82 ± 0.35	0.81 ± 0.37	0.379
eGFR (mL/min/1.73 m^2^)	54	79 ± 33	81 ± 34	0.186
Uric acid (mg/dL)	48	5.9 ± 1.6	5.5 ± 1.9	0.036
Hemoglobin (g/dL)	54	14.4± 1.6	14.5 ± 1.4	0.736
Platelet (×10^4^/μL)	54	23.5 ± 6.8	23.9 ± 6.7	0.213
UACR (mg/g Cre)	36	164 ± 344	119 ± 221	0.950
**(c) The changes in metabolic parameters 12 months after the initiation of semaglutide treatment (n = 40).**				
	**n**	**Baseline**	**12 Months**	** *p* **
Body weight (kg)	32	86.2 ± 18.3	84.5 ± 19.2	0.001
BMI (kg/m^2^)	32	32.6 ± 5.5	31.9 ± 5.8	0.001
Systolic blood pressure (mmHg)	32	134 ± 15	134 ± 18	0.873
Diastolic blood pressure (mmHg)	32	77 ± 10	76 ± 13	0.731
Plasma glucose (mg/dL)	40	175 ± 55	166 ± 59	0.458
HbA1c (%)	40	8.2 ± 1.4	7.4 ± 1.2	<0.001
Albumin (g/dL)	37	4.06 ± 0.51	4.15 ± 0.42	0.110
AST (IU/L)	40	42 ± 32	32 ± 21	0.018
ALT (IU/L)	40	54 ± 40	41 ± 30	0.002
GGTP (IU/L)	38	67 ± 64	57 ± 55	0.071
TC (mg/dL)	37	179 ± 33	171 ± 26	0.113
HDL-C (mg/dL)	39	45 ± 10	48 ± 11	0.034
LDL-C (mg/dL)	36	99 ± 31	96 ± 25	0.539
TG (mg/dL)	39	6.1 ± 3.9	4.8 ± 3.5	0.034
TG/HDL-C	40	242 ± 115	198 ± 110	0.024
Non-HDL-C (mg/dL)	36	135 ± 31	124 ± 27	0.038
Creatinine (mg/dL)	40	0.79 ± 0.31	0.79 ± 0.32	0.835
eGFR (mL/min/1.73 m^2^)	40	81 ± 35	82 ± 36	0.803
Uric acid (mg/dL)	36	5.9 ± 1.6	5.7 ± 1.5	0.139
Hemoglobin (g/dL)	40	14.3± 1.8	14.2 ± 1.6	0.610
Platelet (×10^4^/μL)	40	24.0 ± 6.8	23.9 ± 7.2	0.814
UACR (mg/g Cre)	28	201 ± 381	138 ± 289	0.221

Values show mean ± SD. ALT, alanine aminotransferase; AST, aspartate aminotransferase; BMI, body mass index; eGFR, estimated glomerular filtration rate; GGTP, gamma-glutamyl transferase; HbA1c, hemoglobin A1c; HDL-C, high-density lipoprotein cholesterol; LDL-C, low-density lipoprotein cholesterol; TC, total cholesterol; TG, triglyceride; UACR, albumin-to-creatinine ratio.

**Table 3 biomedicines-12-01001-t003:** The changes in metabolic parameters during the 12-month semaglutide treatment in patients who were GLP-1RA naïve or given semaglutide switched from other GLP-1RAs.

	GLP-1RA Naïve (n = 19)				Switch from Other GLP-1RAs (n = 21)			
	n	Baseline	12 Months	*p*	n	Baseline	12 Months	*p*
Body weight (kg)	14	91.2 ± 23.2	89.2 ± 24.1	0.076	18	82.3 ± 11.8	80.9 ± 13.1	0.053
BMI (kg/m^2^)	14	34.1 ± 6.4	33.3 ± 6.7	0.070	18	31.4 ± 4.3	30.9 ± 4.7	0.051
Systolic blood pressure (mmHg)	14	135 ± 14	132 ± 23	0.701	18	132 ± 15	136 ± 12	0.311
Diastolic blood pressure (mmHg)	14	80 ± 11	79 ± 18	0.867	18	75 ± 8	74 ± 8	0.673
Plasma glucose (mg/dL)	19	191 ± 59	153 ± 38	0.004	21	161 ± 47	178 ± 70	0.333
HbA1c (%)	19	8.4 ± 1.3	7.1 ± 0.9	<0.001	21	8.2 ± 1.4	7.7 ± 1.4	0.032
Albumin (g/dL)	19	4.16 ± 0.24	4.20 ± 0.48	0.482	18	3.95 ± 0.66	4.10 ± 0.53	0.157
AST (IU/L)	19	50 ± 30	33 ± 19	0.006	21	34 ± 33	30 ± 24	0.868
ALT (IU/L)	19	67 ± 44	43 ± 29	0.003	21	42 ± 31	38 ± 30	0.287
GGTP (IU/L)	18	88 ± 73	62 ± 49	0.008	20	48 ± 47	52 ± 60	0.762
TC (mg/dL)	18	192 ± 33	173 ± 28	0.021	19	166 ± 27	169 ± 24	0.614
HDL-C (mg/dL)	19	47 ± 9	49 ± 10	0.356	20	42 ± 9	46 ± 12	0.038
LDL-C (mg/dL)	16	106 ± 35	90 ± 25	0.067	20	93 ± 26	100 ± 25	0.218
TG (mg/dL)	19	244 ± 10	209 ± 112	0.212	21	241 ± 123	188 ± 107	0.085
TG/HDL-C	19	5.5 ± 3.2	4.5 ± 2.7	0.136	20	6.6 ± 4.6	5.1 ± 4.3	0.100
Non-HDL-C (mg/dL)	18	145 ± 33	125 ± 28	0.019	20	126 ± 26	124 ± 27	0.793
Creatinine (mg/dL)	19	0.76 ± 0.25	0.77 ± 0.26	0.716	21	0.82 ± 0.35	0.80 ± 0.37	0.266
eGFR (mL/min/1.73 m^2^)	19	78 ± 22	76 ± 23	0.798	21	84 ± 44	86 ± 44	0.346
Uric acid (mg/dL)	17	5.8 ± 1.4	5.3 ± 1.4	0.746	19	6.1 ± 1.7	6.0 ± 1.5	0.731
Hemoglobin (g/dL)	19	14.6 ± 1.2	14.3 ± 0.8	0.393	21	14.0 ± 2.1	14.0 ± 2.1	0.961
Platelet (×10^4^/μL)	19	24.5 ± 6.7	25.3 ± 7.1	0.573	21	23.6 ± 6.9	22.6 ± 7.1	0.263
UACR (mg/g Cre)	13	86 ± 104	53 ± 74	0.463	15	300 ± 491	212 ± 373	0.281

Values show mean ± SD. ALT, alanine aminotransferase; AST, aspartate aminotransferase; BMI, body mass index; eGFR, estimated glomerular filtration rate; GGTP, gamma-glutamyl transferase; GLP-1RA, glucagon-like peptide-1 receptor agonist; HbA1c, hemoglobin A1c; HDL-C, high-density lipoprotein cholesterol; LDL-C, low-density lipoprotein cholesterol; TC, total cholesterol; TG, triglyceride; UACR, albumin-to-creatinine ratio.

**Table 4 biomedicines-12-01001-t004:** The changes in the indices of MASLD during the 12-month semaglutide treatment.

**(a) The changes in the indices of MASLD in all patients (n = 40).**								
	**n**	**Baseline**	**12 Months**	** *p* **				
HSI	40	39.6 ± 14.7	38.6 ± 13.9	0.051				
APRI	40	0.485 ± 0.421	0.374 ± 0.298	0.288				
FIB-4 index	36	1.50 ± 1.12	1.37 ± 0.80	0.765				
**(b) The changes in the indices of MASLD in patients who were GLP-1RA naïve or (n = 19) or given semaglutide after being switched from other GLP-1RAs (n = 21).**								
			**Group A** **GLP-1RA Naïve (n = 19)**			**Group B** **Switch from GLP-1RAs (n = 21)**		
	n	Baseline	12 months	*p*	n	Baseline	12 months	*p*
HSI	19	38.8 ± 16.3	37.7 ± 15.7	0.145	21	40.4 ± 12.9	39.4 ± 12.0	0.205
APRI	19	0.591 ± 0.476	0.360 ± 0.207	0.016	21	0.390 ± 0.227	0.387 ± 0.261	0.217
FIB-4 index	19	1.63 ± 1.42	1.29 ± 0.85	0.117	17	1.36 ± 0.58	1.46 ± 0.73	0.103
**(c) The changes in the indices of MASLD in patients with or without a high degree of liver damage at baseline (ALT ≥ 30 (n = 20) or <30 (n = 20)).**								
			**Baseline ALT ≥ 30 (n = 20)**			**Baseline ALT < 30 (n = 20)**		
	n	Baseline	12 months	*p*	n	Baseline	12 months	*p*
HSI	20	40.6 ± 16.0	39.7 ± 13.7	0.314	20	38.7 ± 13.2	37.4 ± 12.0	0.108
APRI	20	0.738 ± 0.461	0.477 ± 0.202	0.010	20	0.233 ± 0.122	0.272 ± 0.182	0.005
FIB-4 index	19	1.77 ± 1.37	1.34 ± 0.77	0.033	17	1.20 ± 0.62	1.40 ± 0.84	0.009
**(d) The changes in the indices of MASLD in patients divided by baseline LDL-C values (LDL-C ≥ 100 (n = 22) or <100 (n = 18)).**								
			**Baseline LDL-C** ≥ **100 (n = 22)**			**Baseline LDL-C < 100 (n = 18)**		
	n	Baseline	12 months	*p*	n	Baseline	12 months	*p*
HSI	22	41.2 ± 14.3	39.2 ± 13.2	0.010	18	37.7 ± 15.7	37.8 ± 15.5	0.913
APRI	22	0.483 ± 0.395	0.384 ± 0.368	0.355	18	0.489 ± 0.473	0.362 ± 0.205	0.586
FIB-4 index	21	1.28 ± 0.78	1.26 ± 0.88	0.794	15	1.81 ± 1.47	1.53 ± 0.71	0.820
**(e) The changes in the indices of MASLD in patients who were treated with or without statins.**								
			**With Statins (n = 21)**			**Without Statins (n = 19)**		
	n	Baseline	12 months	*p*	n	Baseline	12 months	*p*
HSI	21	36.9 ± 16.2	35.4 ± 14.9	0.030	19	42.7 ± 13.0	42.1 ± 12.5	0.520
APRI	21	0.496 ± 0.447	0.409 ± 0.370	0.715	19	0.473 ± 0.414	0.336 ± 0.206	0.198
FIB-4 index	18	1.72 ± 1.43	1.55 ± 0.93	0.879	18	1.29 ± 0.70	1.20 ± 0.66	0.744
**(f) The changes in the indices of MASLD in patients who were treated with or without ARB.**								
			**With ARB (n = 23)**			**Without ARB (n = 17)**		
	n	Baseline	12 months	*p*	n	Baseline	12 months	*p*
HSI	23	36.0 ± 14.0	34.5 ± 13.1	0.033	17	44.6 ± 14.9	44.2 ± 13.7	0.653
APRI	23	0.470 ± 0.450	0.374 ± 0.355	0.831	17	0.506 ± 0.406	0.374 ± 0.221	0.227
FIB-4 index	21	1.72 ± 1.40	1.54 ± 0.89	0.958	17	1.19 ± 0.50	1.13 ± 0.64	0.820

Values show mean ± SD. APRI, AST-to-platelet ratio index; AST, aspartate aminotransferase; ARB, angiotensin II receptor blocker; AST, aspartate aminotransferase; FIB-4 index, fibrosis-4 index; GLP-1RA, glucagon-like peptide-1 receptor agonist; HbA1c, hemoglobin A1c; HDL-C, high-density lipoprotein cholesterol; HSI, hepatic steatosis index; LDL-C, low-density lipoprotein cholesterol; MASLD, metabolic dysfunction-associated steatotic liver disease.

**Table 5 biomedicines-12-01001-t005:** Spearman’s rank correlation coefficient among the changes in metabolic parameters during the 12-month semaglutide treatment. (* *p* < 0.05, ** *p* < 0.01).

	Δ BMI	Δ HbA1c	Δ TG	Δ HDL-C	Δ LDL-C	ΔTG/ HDL-C	Δ Non- HDL-C	Δ HSI	Δ APRI
Δ BMI	1								
Δ HbA1c	0.146	1							
Δ TG	0.030	0.124	1						
Δ HDL-C	−0.084	−0.026	−0.356 *	1					
Δ LDL-C	0.497 **	0.109	0.064	0.035	1				
ΔTG/HDL-C	−0.169	0.409	0.988 **	−0.440	0.006	1			
Δ Non- HDL-C	0.245	0.284	0.467 **	−0.139	0.834 **	0.472	1		
Δ HSI	0.536 **	0.024	0.114	0.211	0.236	−0.254	0.154	1	
ΔAPRI	0.229	0.494 **	0.145	−0.097	0.160	0.412	0.327	−0.061	1
Δ FIB-4 index	0.018	0.511 **	0.196	−0.215	−0.016	0.534 *	0.140	−0.346 *	0.874 **

APRI, AST-to-platelet ratio index; AST, aspartate aminotransferase; BMI, body mass index; FIB-4 index, fibrosis-4 index; HbA1c, hemoglobin A1c; HDL-C, high-density lipoprotein cholesterol; LDL-C, low-density lipoprotein cholesterol; TG, triglyceride.

**Table 6 biomedicines-12-01001-t006:** The changes in the UACR stage in patients who were prescribed semaglutide for 12 months.

UACR Stage at Baseline	n	UACR Stage at 12 Months	n
A1	13	A1 A2	12 1
A2	10	A1 A2	3 7
A3	5	A2 A3	2 3

UACR, albumin-to-creatinine ratio.

## Data Availability

The data supporting the findings of this study are available from the corresponding author upon reasonable request.

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
