# Peer review of "Effects of Once-Weekly Semaglutide on Cardiovascular Risk Factors and Metabolic Dysfunction-Associated Steatotic Liver Disease in Japanese Patients with Type 2 Diabetes: A Retrospective Longitudinal Study Based on Real-World Data"

_biomedicines, 2024, doi:10.3390/biomedicines12051001_

Round 1

Reviewer 1 Report

Comments and Suggestions for Authors

In Katsuyama et al.'s study, a retrospective analysis of 75 patients with T2D who were prescribed once-weekly semaglutide was conducted to compare metabolic parameters before and after the initiation of semaglutide. Significant decreases in body weight, HbA1c, and non-HDL cholesterol were observed at 3, 6, and 12 months post-initiation, with HDL-cholesterol showing an increase at 12 months. Furthermore, during the 12-month semaglutide treatment period, a significant decrease in the hepatic steatosis index (HSI) was noted, along with a similar trend observed in the AST to platelet ratio index (APRI). Consequently, this real-world study confirms the beneficial effects of once-weekly semaglutide in improving body weight, glycemic control, and the atherogenic lipid profile, as well as its positive effects on MASLD.

Before proceeding, it is important to address the following comments:

Major comments:

1)      In addition to pharmacological approaches, non-pharmacological interventions are crucial for improving metabolic parameters. It would be valuable to provide information on the status of other interventions, such as adopting a healthy diet, smoking cessation, and engaging in physical activity.

2)      Evaluating the ratio of TG/HDL would provide insightful information on insulin resistance during semaglutide therapy (Table 3) and its correlation with other metabolic parameters shown in Table 5.

3)      Although significance was not found in the absolute values of UACR, it would be valuable to inform what percentage of individuals experienced changes in UACR stage, for instance, from A3 to A2 or A1, and from A2 to A1. Likewise, the analysis of eGFR slope during semaglutide treatment should also be addressed. Make ammendments to the manuscript accordingly. 

Minor comments:

1)      Have amylase and lipase levels increased in any patient during treatment with semaglutide? While these laboratory changes are rare, they necessitate drug withdrawal when they occur.

2)      Why were only 50% of patients receiving blockers of the renin-angiotensin-aldosterone system? This class of medication is widely used for renal and cardiovascular protection, and the lower rate of its use warrants further explanation.  

Author Response

In Katsuyama et al.'s study, a retrospective analysis of 75 patients with T2D who were prescribed once-weekly semaglutide was conducted to compare metabolic parameters before and after the initiation of semaglutide. Significant decreases in body weight, HbA1c, and non-HDL cholesterol were observed at 3, 6, and 12 months post-initiation, with HDL-cholesterol showing an increase at 12 months. Furthermore, during the 12-month semaglutide treatment period, a significant decrease in the hepatic steatosis index (HSI) was noted, along with a similar trend observed in the AST to platelet ratio index (APRI). Consequently, this real-world study confirms the beneficial effects of once-weekly semaglutide in improving body weight, glycemic control, and the atherogenic lipid profile, as well as its positive effects on MASLD.

Before proceeding, it is important to address the following comments:

Major comments:

  • In addition to pharmacological approaches, non-pharmacological interventions are crucial for improving metabolic parameters. It would be valuable to provide information on the status of other interventions, such as adopting a healthy diet, smoking cessation, and engaging in physical activity.

We appreciate the reviewer’s suggestion. Appropriate diet/physical activity is important and can have a great impact on metabolic disorders. All patients were under standard care including recommendations for diet and physical activities according to the guidelines of the Japan Diabetes Society. However, evaluations of lifestyle were difficult due to the nature of a retrospective study. We added a description of it in “Results” and “Discussion”.

  • Evaluating the ratio of TG/HDL would provide insightful information on insulin resistance during semaglutide therapy (Table 3) and its correlation with other metabolic parameters shown in Table 5.

We evaluated the ratio of TG/HDL-C and showed the results in Tables 2, 3 and 5.

3)      Although significance was not found in the absolute values of UACR, it would be valuable to inform what percentage of individuals experienced changes in UACR stage, for instance, from A3 to A2 or A1, and from A2 to A1. Likewise, the analysis of eGFR slope during semaglutide treatment should also be addressed. Make ammendments to the manuscript accordingly.

We evaluated the transition of the UACR stage during the 12-month semaglutide treatment and showed the results in Table 6.

Minor comments:

  • Have amylase and lipase levels increased in any patient during treatment with semaglutide? While these laboratory changes are rare, they necessitate drug withdrawal when they occur.

Since the regular measurements of amylase and lipase are not common, it is impossible to show these values in this study.

2)      Why were only 50% of patients receiving blockers of the renin-angiotensin-aldosterone system? This class of medication is widely used for renal and cardiovascular protection, and the lower rate of its use warrants further explanation.  

In Japan, blockers of the renin-angiotensin-aldosterone system are indicated for the treatment of hypertension. In our study, patients without hypertension were also included. The rate of patients who were given these agents was about 50 % in our previous study. We presume this reflects the real-world situation.

Reviewer 2 Report

Comments and Suggestions for Authors

The authors studied the efficacy of once-weekly semaglutide in the real-world application. Retrospective collection of data from the 75 Japanese patients with T2D was performed from 2020 to 2022. Several metabolic parameters were checked before and after the administration of semaglutide given at 3, 6 and 12 months. In addition, effects on HIS, APRI, and MASLD were analyzed. The authors then conclude that in their real-world study once-weekly semaglutide improved body weight, glycemic control and atherogenic lipidemia in Japanese patients with T2D. Moreover, it was also suggested that semaglutide reduces hepatic fat content and has a preventing role against the progression of MASLD in such patient populations.

Minor comments:

Page 10, line 216: “52-month” ? From the reference 11: 56 weeks and the reference 10: 30 weeks, respectively

Page 10, line 224: If the present study results are similar to those reported in the reference 21, it would be necessary to explain the reason to repeat such study (e.g., real-world study, Japanese, once -weekly semaglutide)  

Page 11, line 295: Although you have pointed out several limitations of this retrospective study, how far your results and conclusion can be reached among yours and those in literatures (e.g., data from prior large-scale clinical trials)? Namely, the differences and generalizability of this study should be commented.  

Author Response

The authors studied the efficacy of once-weekly semaglutide in the real-world application. Retrospective collection of data from the 75 Japanese patients with T2D was performed from 2020 to 2022. Several metabolic parameters were checked before and after the administration of semaglutide given at 3, 6 and 12 months. In addition, effects on HIS, APRI, and MASLD were analyzed. The authors then conclude that in their real-world study once-weekly semaglutide improved body weight, glycemic control and atherogenic lipidemia in Japanese patients with T2D. Moreover, it was also suggested that semaglutide reduces hepatic fat content and has a preventing role against the progression of MASLD in such patient populations.

Minor comments:

Page 10, line 216: “52-month” ? From the reference 11: 56 weeks and the reference 10: 30 weeks, respectively

We appreciate the reviewer’s comment. “52-month” was our mistake. We omitted the description of the treatment period.

Page 10, line 224: If the present study results are similar to those reported in the reference 21, it would be necessary to explain the reason to repeat such study (e.g., real-world study, Japanese, once -weekly semaglutide)  

The previous study focused only on body weight and HbA1c which differed from our study. We added a description of the point in “Discussion”.

Page 11, line 295: Although you have pointed out several limitations of this retrospective study, how far your results and conclusion can be reached among yours and those in literatures (e.g., data from prior large-scale clinical trials)? Namely, the differences and generalizability of this study should be commented. 

We considered that our study confirmed the results of previous clinical trials showing that once-weekly semaglutide on body weight, HbA1c and atherogenic lipid profiles, which enables it to be generalized to real-world patients with T2D. We described this point in the last part of “Discussion”.

Reviewer 3 Report

Comments and Suggestions for Authors

Authors studied the potential role of semaglutide on liver steatosis of t2dm patients in Japan. The work confirms previous data western patients and suggest this interesting antidiabetic drug for liver diseases. However, the number of patients is low, and some data should be reconsidered before taking conclusions.  

Mainly, data from the liver markers of damage must be adjusted by pharmacology. Many patients also took metformin, sglt2i, statin and ARA2. Thus, improvements may come from these other drugs rather than from glp1ra (or by additive effects)

Also, these data can be adjusted by lipid levels since these molecules may mediate liver alterations.

A HOMA-IR or OGTT data could add interesting data about t2dm progression.

Some tables (in Table 2) should be better shown in graph.

Also, comparative tables (in Table 3) on glp1ra naïve and switched from another glp1ra could be avoided. Only significant data were found mainly in HbA1ac. Remarkedly, those patients switched from another glp1ra from slimer than the others. Thus, significant differences between them should be adjusted by BMI or body weight.

Semaglutide can be stronger anti-hyperglycemic drug than dulaglutide? Please, discuss

Legend in Table 5 must be completed with details of figure and statistic.

The UCAR data was obtained in very few patients. Please, reconsider to delete.

Comments on the Quality of English Language

Fine

Reviewer 4 Report

Comments and Suggestions for Authors

I’ve read with attention the paper of Katsuyama et al. that is potentially of interest. The background and aim of the study have been clearly defined. The methodology applied is overall correct, but it can be improved. The reported results are reliable and adequately discussed. The conclusions are consistent with the evidence and arguments presented and they address the main question posed. The references are also appropriate as well as tables and figures. I have no ethical concerns regarding experiments, nor on plagiarism or publication ethics. I’ve however some comments:

- The abstract should contain some (main) quantitative data

- Are HSI, APRI and FIB-4 markers of MASLD? They were validated before the MASLD definition and they were more specifically related to the liver steatosis and fibrosis stage. This has to be more deeply discussed. Moreover, it is not clear why the authors choose the HSI and not the Fatty Liver Index, that is more validated and used in literature

- From a statistical point of view, the authors managed all data as they were all normally distributed. It is hardly believable that all the investigated parameters are normally distributed, but the authors also did not test the parameters distribution. This should be ammended.

Comments on the Quality of English Language

Overall fine. Some sentences are long.

Author Response

I’ve read with attention the paper of Katsuyama et al. that is potentially of interest. The background and aim of the study have been clearly defined. The methodology applied is overall correct, but it can be improved. The reported results are reliable and adequately discussed. The conclusions are consistent with the evidence and arguments presented and they address the main question posed. The references are also appropriate as well as tables and figures. I have no ethical concerns regarding experiments, nor on plagiarism or publication ethics. I’ve however some comments:

- The abstract should contain some (main) quantitative data

We appreciate the reviewer’s suggestion. We added the changes in body weight and HbA1c in “Abstract”.

- Are HSI, APRI and FIB-4 markers of MASLD? They were validated before the MASLD definition and they were more specifically related to the liver steatosis and fibrosis stage. This has to be more deeply discussed. Moreover, it is not clear why the authors choose the HSI and not the Fatty Liver Index, that is more validated and used in literature

Although HSI, APRI and FIB-4 index were well validated for the evaluation of NAFLD in previous reports, there are few reports assessing the association between these indexes and the severity or outcome of newly defined MASLD. We also considered evaluating fatty liver index. However, it was impossible in this retrospective study, since the regular measurements of waist circumference were not common in clinical practice in Japan. We added these points to “Discussion”.

- From a statistical point of view, the authors managed all data as they were all normally distributed. It is hardly believable that all the investigated parameters are normally distributed, but the authors also did not test the parameters distribution. This should be ammended.

We tested the distributions in all values using Shapiro–Wilk test and used Wilcoxon signed-rank test for comparisons of the variables without normal distribution. Correlations were tested using Spearman’s rank correlation coefficient instead of Pearson’s correlation coefficient.

Reviewer 5 Report

Comments and Suggestions for Authors

After carefully reviewing the manuscript entitled 'Effects of Semaglutide administered once weekly on cardiovascular risk factors and steatotic liver disease associated with metabolic dysfunction in Japanese patients with type 2 diabetes: a longitudinal retrospective study based on real-world data', I am compelled to recommend its rejection for publication on the following reasons:

Plagiarism: There's plagiarism, check the report provided by mdpi

Sample size: The study is based on a sample of 75 patients, which appears too small for a retrospective investigation aimed at exploring complex effects such as those mentioned. This sample size limits the generalisability of the results and their statistical reliability.

Confounding factors not assessed: Despite the aim of examining the effects of semaglutide, the study does not properly account for important confounding variables such as participants' diet and physical activity. This omission may significantly influence the results, as these factors play a crucial role in modulating cardiovascular risks and the progression of steatotic liver disease.

Clarity of the study design: The study design and methods used are superficially described and do not provide sufficient detail to enable a complete understanding of the research conducted. Specifically, there is a lack of clear information on the selection of participants, the handling of missing data and statistical analysis. This makes it difficult to assess the appropriateness of the procedures adopted and the validity of the conclusions drawn.

Comments on the Quality of English Language

English is fine

Author Response

After carefully reviewing the manuscript entitled 'Effects of Semaglutide administered once weekly on cardiovascular risk factors and steatotic liver disease associated with metabolic dysfunction in Japanese patients with type 2 diabetes: a longitudinal retrospective study based on real-world data', I am compelled to recommend its rejection for publication on the following reasons:

Plagiarism: There's plagiarism, check the report provided by mdpi

This report did not include plagiarism. We had already conducted a retrospective study evaluating the real-world efficacy of GLP-1RA dulaglutide. In the current study, we examined the effects of once-weekly semaglutide, the other GLP-1RA, which had been suggested to have stronger effects on various metabolic factors. This study was completely new with different data. As the reviewer can recognize, the patient characteristics as well as the results were completely different from the former study. Nevertheless, the study design of the current study resembled that of the former study, which might cause a high repetition rate. During the revision, we have tried to reduce the repetition as much as possible.

Sample size: The study is based on a sample of 75 patients, which appears too small for a retrospective investigation aimed at exploring complex effects such as those mentioned. This sample size limits the generalisability of the results and their statistical reliability.

As the reviewer pointed out, the limited number of patients was one of the limitations of our study. We discussed this point in “Discussion”. In clinical practice, semaglutide is mainly prescribed in patients with T2D with a high risk for cardiovascular diseases. In such patients, a number of medications such as SGLT2is, metformin, ARB and statins, were already prescribed before the initiation of GLP-1RAs. Patients, who were treated only with GLP-1RAs are extremely rare. Our investigation reflected the situation of the real-world, which was different from the clinical trials.

Confounding factors not assessed: Despite the aim of examining the effects of semaglutide, the study does not properly account for important confounding variables such as participants' diet and physical activity. This omission may significantly influence the results, as these factors play a crucial role in modulating cardiovascular risks and the progression of steatotic liver disease.

All patients were instructed to keep diet and physical activity from the guidelines of the Japan Diabetes Society. Nevertheless, it was difficult to evaluate the quality and situation of diet and physical activity in each patient in such a retrospective study. We discussed the limitation in “Discussion”.

Clarity of the study design: The study design and methods used are superficially described and do not provide sufficient detail to enable a complete understanding of the research conducted. Specifically, there is a lack of clear information on the selection of participants, the handling of missing data and statistical analysis. This makes it difficult to assess the appropriateness of the procedures adopted and the validity of the conclusions drawn.

We appreciate the suggestion. We tried to improve our statistical analysis and revise the description in “Methods”.

Round 2

Reviewer 1 Report

Comments and Suggestions for Authors

I have no more comments. Congratulations on your work.

Author Response

We cordially appreciate again for the review, which contributed to improving our report.

Reviewer 3 Report

Comments and Suggestions for Authors

Thanks for the answers. 

We appreciate the reviewer’s suggestion. As the reviewer pointed out, the results of our study could be influenced by concomitant agents such as SGLT2is, metformin statins, and ARA2.

This should be clearly atated as a limitation of the study. Nevertheless, the population can be equally divided between patients with/out ARA2 or statin. Please, discuss data after adjustements

As the reviewer’s the suggestion, there were close associations between lipid profiles and liver function. Nevertheless, it was difficult to choose the specific lipoprotein and decide its cutoff point for the adjustments.

Please, test the main lipoproteins such as LDL-C at classical cut-off for these patients

Regular measurements of insulin levels or OGTT are not common in clinical practice in Japan. Thus, it’s impossible to add the data.

We would like to show that there were reductions in body weight and HbA1c even in patients who were given semaglutide switched from other GLP-1RAs. Thus, we left the data.

Please, avoid more tables than needed. Include it as a text data

Author Response

This should be clearly atated as a limitation of the study. Nevertheless, the population can be equally divided between patients with/out ARA2 or statin. Please, discuss data after adjustements

We cordially appreciate for the review, which contributed to improving our report. We added Table 4 (e-f), which shows the comparisons of the MASLD indexes in patients with/without statins or ARB. We also discussed the possible influences of the use of these drugs in “Discussion”.

Please, test the main lipoproteins such as LDL-C at classical cut-off for these patients

In Table 4 (d), we showed the changes in MASLD indexes in patients divided by the baseline LDL-C levels.

Please, avoid more tables than needed. Include it as a text data

We appreciate for the suggestion. As the reviewer pointed out, unnecessary data should be avoided. Nevertheless, the comparisons between GLP-1RA naïve patients and those switched from other GLP-1RAs are one of the most important parts of this study. In “Discussion”, we discussed the comparisons between semaglutide and other GLP-1RAs based on these data. Even if there were lacks of significant differences in most of values, it is impossible for us ignore the importance of the data. We are grateful for your understanding.

Reviewer 4 Report

Comments and Suggestions for Authors

The authors have considered the reviewers' suggestions and improved the paper accordingly. I have no further comments on it.

Author Response

(The authors gave the same response as above.)

Reviewer 5 Report

Comments and Suggestions for Authors

The authors have made some improvements to the paper. The critical points pointed out in the previous review remain, especially the too small sample for a retrospective study and the lack of analysis of factors such as diet are important limitations of the paper. 

Comments on the Quality of English Language

English is fine

Author Response

The authors have made some improvements to the paper. The critical points pointed out in the previous review remain, especially the too small sample for a retrospective study and the lack of analysis of factors such as diet are important limitations of the paper.

We cordially appreciate the review, which contributed to improving our report. As the reviewer pointed out, the most important limitations of our study are the lacks of evaluations on diet and physical activities as well as the small number of patients. Nevertheless, it is difficult to increase the number of patients and add further information on lifestyles due to data availability. Instead, we changed the descriptions of the “Discussion” to emphasize these limitations.

Round 3

Reviewer 5 Report

Comments and Suggestions for Authors

The critical issues highlighted in the previous review still stand, particularly the insufficient sample size for a retrospective study and the omission of factor analysis such as diet, which represent significant limitations of the paper.

Comments on the Quality of English Language

English is fine

Author Response

The critical issues highlighted in the previous review still stand, particularly the insufficient sample size for a retrospective study and the omission of factor analysis such as diet, which represent significant limitations of the paper.

We cordially appreciate the review. Further analysis is currently difficult because of the data availability. We modified the descriptions in “Discussion” to emphasize our limitations.